# Physiological and Transcriptome Analyses Reveal the Protective Effect of Exogenous Trehalose in Response to Heat Stress in Tea Plant (*Camellia sinensis*)

**DOI:** 10.3390/plants13101339

**Published:** 2024-05-13

**Authors:** Shizhong Zheng, Chufei Liu, Ziwei Zhou, Liyi Xu, Zhongxiong Lai

**Affiliations:** 1College of Biological Science and Engineering, Ningde Normal University, Ningde 352100, China; zhengshizhong@126.com (S.Z.); chufei0626@126.com (C.L.); zwchow92@126.com (Z.Z.); xuliyi1990@outlook.com (L.X.); 2Institute of Horticultural Biotechnology, Fujian Agriculture and Forestry University, Fuzhou 350002, China

**Keywords:** trehalose, *Camellia sinensis*, heat stress, physiological analysis, transcriptome analysis

## Abstract

It is well known that application of exogenous trehalose can enhance the heat resistance of plants. To investigate the underlying molecular mechanisms by which exogenous trehalose induces heat resistance in *C. sinensis*, a combination of physiological and transcriptome analyses was conducted. The findings revealed a significant increase in the activity of superoxide dismutase (SOD) and peroxidase (POD) upon treatment with 5.0 mM trehalose at different time points. Moreover, the contents of proline (PRO), endogenous trehalose, and soluble sugar exhibited a significant increase, while malondialdehyde (MDA) content decreased following treatment with 5.0 mM trehalose under 24 h high-temperature stress (38 °C/29 °C, 12 h/12 h). RNA-seq analysis demonstrated that the differentially expressed genes (DEGs) were significantly enriched in the MAPK pathway, plant hormone signal transduction, phenylpropanoid biosynthesis, flavone and flavonol biosynthesis, flavonoid biosynthesis, and the galactose metabolism pathway. The capability to scavenge free radicals was enhanced, and the expression of a heat shock factor gene (*HSFB2B*) and two heat shock protein genes (*HSP18.1* and *HSP26.5*) were upregulated in the tea plant. Consequently, it was concluded that exogenous trehalose contributes to alleviating heat stress in *C. sinensis*. Furthermore, it regulates the expression of genes involved in diverse pathways crucial for *C. sinensis* under heat-stress conditions. These findings provide novel insights into the molecular mechanisms underlying the alleviation of heat stress in *C. sinensis* with trehalose.

## 1. Introduction

The tea plant (*Camellia sinensis*) is an important non-alcoholic beverage crop, originating in southwest China [1]. It thrives in environments with specific temperature and water requirements. Ideally, temperatures ranging from 20 to 25 °C provide the most suitable condition for tea plant growth. However, when temperature exceed 30 °C, the growth rate of tea plants significantly decreases. Prolonged exposure to temperature exceeding 35 °C for 10–15 consecutive days can lead to the withering and eventual death of tea plants [2]. With the ongoing challenge of global warming, the frequency of extreme high-temperature events during summer is on the rise, impacting the yield and quality of tea [3]. High temperature not only affects the physical appearance of plants but also disrupts various physiological and biochemical processes. It has been reported that constantly elevated temperature can result in leaf discoloration, curling and senescence, leaf and twig scorching, inhibited stem growth, and reduced plant height [4,5,6]. Furthermore, high temperatures can disrupt the photosynthesis of tea plants by deactivating photosystems, light-harvesting chlorophyll protein complexes, and altering carbon fixation [7]. In addition, high temperature leads to excessive accumulation of reactive oxygen species (ROS) and malondialdehyde (MDA), causing structural damage to plant cells [8]. Therefore, it is of great significance to develop strategies to improve the heat-stress resistance of tea plants.

Trehalose, a non-reducing disaccharide, composed of two glucose molecules, serves several functional roles, such as conferring thermostability, providing osmotic protection, storing energy, regulating growth, and participating in sugar signaling [9]. Trehalose metabolism has been associated with cell survival under adverse conditions, making it crucial for stress tolerance in vivo [10,11,12,13]. Exogenous trehalose application enhances a plant’s ability to withstand stress related to high temperature. For example, a concentration of trehalose (50 mmol/L) increased the survival rate of *Saccharomyces cerevisiae* at 53 °C possibly by reducing ROS levels and lipid peroxidation damage [14]. Trehalose has also been found to regulate the heat-stress response in *Pleurotus ostreatus* by influencing central carbon metabolism [15]. Moreover, trehalose could also improve the heat resistance of *Lentinula edodes* mycelia [16]. In maize and wheat, trehalose has been shown to alleviate the decrease in the net photosynthetic rate at 42 °C, increase the activity of ribulose-1,5-bisphosphate carboxylase, and promote the Calvin cycle [17]. Furthermore, trehalose can alleviate high-temperature damage by promoting photosynthetic electron transport, protecting proteins, reducing chloroplasts damage, and enhancing antioxidant systems in maize, wheat, and peony [18,19,20,21]. However, the biological function of trehalose in *C. sinensis*, particularly its role in enhancing heat tolerance, remains poorly understood.

Based on these findings and the effects of trehalose on the heat resistance in various plant species, we hypothesized that trehalose could protect tea plants from heat stress. To clarify the specific effect of trehalose on the heat resistance of tea plants, this study analyzed the physiological and morphological characteristics of tea plants under heat stress with exogenous trehalose treatment. Meanwhile, RNA-Seq technology was employed to explore the specific genes involved in regulatory mechanism of trehalose-mediated heat response in tea plants. Our analyses lay the groundwork for understanding the function of trehalose in the heat response in the tea plant, potentially providing insights to improve tea leaves’ yield and quality.

## 2. Results

### 2.1. Effects of Exogenous Trehalose on Growth Morphology of Tea Plant under Heat Stress

As depicted in Figure 1A, tea plant leaves of each group were subjected to a 2 d pretreatment of heat stress, followed by rolling. Subsequently, the leaves were sprayed with water and varying concentrations of trehalose (2.5 mM, 5.0 mM, and 10.0 mM) (Figure 1A). The effect of exogenous trehalose on the growth morphology of tea plants under heat stress is illustrated in Figure 1B. Compared with the control group, the 2.5 mM trehalose treatment group showed no discernible difference in growth morphology between the two groups at 0 h and 12 h. By 24 h, the leaves of the control group began to wilt, while those treated with 2.5 mM trehalose showed mild scorching and an upturned appearance. At 36 h, both groups displayed wilting, with the control group exhibiting more severe wilting. By 48 h, the tea plants in the control group showed yellowing and severe wilting, while the leaves treated with 2.5 mM trehalose were just severely wilted (Figure 1B).

To explore the effects of different concentrations of trehalose, we also evaluated the growth difference of tea plants treated with 2.5 mM, 5.0 mM, and 10.0 mM trehalose. As shown in Figure 1B, there was no difference in the growth morphology between the 2.5 mM and 5.0 mM groups at 0 h and 12 h. By 24 h, the tea plants treated with 5.0 mM trehalose exhibited better growth status than those treated with 2.5 mM trehalose. At 36 h and 48 h, the tea plants treated with 5.0 mM trehalose showed slight wilting but maintained a healthier overall appearance than those treated with 2.5 mM trehalose. Comparisons between tea plants treated with 10.0 mM and 5.0 mM trehalose revealed that the base of leaves treated with 10.0 mM trehalose showed signs of scorching at 12 h. Subsequently, at 24 h, 36 h, and 48 h, the group treated with 10.0 mM trehalose showed mild, moderate, and severe scorched leaves, respectively. Throughout the process of high-temperature treatment process, 5.0 mM trehalose proved more effective than 2.5 mM and 10.0 mM trehalose in alleviating the heat stress on tea plants. Overall, our results suggest that the application of 5.0 mM trehalose is more efficient in alleviating the damage caused by high temperatures on the growth morphology of tea plants.

### 2.2. Effects of Exogenous Trehalose on Physiological Characteristics of Tea Plant Response to Heat Stress

To clarify the effects of exogenous trehalose on antioxidant enzymes, we analyzed the changes in physiological characteristics under heat-stress conditions. The antioxidant enzyme activity, osmotic substances, endogenous trehalose, and soluble sugar content were measured. As shown in Figure 1C, exogenous trehalose did not influence the SOD activity at 0 and 12 h, but at 24 h, 36 h, and 48 h under heat-stress treatment, the SOD activity of the 5.0 mM trehalose application group was significantly higher than the other groups. During the heat stress process, treatment with 5.0 mM trehalose significantly improved POD activity at 24 h, while POD activity did not show significant changes after treatment with 2.5 mM and 10.0 mM exogenous trehalose (Figure 1D).

To assess the impact of exogenous trehalose on tea plants’ biomembranes, we analyzed the levels of MDA and PRO. After 5.0 mM exogenous trehalose treatment, MDA activity decreased by 16.9%, 17.0%, and 11.7% at 24, 36, and 48 h, respectively (Figure 1E). In comparison to the control, trehalose treatment significantly increased PRO content by 71.0%, 34.9%, 84.9%, and 74.0% under heat stress at 12 h, 24 h, 36 h, and 48 h, respectively (Figure 1F). These results suggest that trehalose treatment alleviated heat-induced oxidative stress in tea leaves. Except for the 0 h group, which was unaffected by exogenous trehalose, the other four treatment groups exhibited that supplementation with exogenous trehalose significantly increased the levels of endogenous trehalose under heat stress (Figure 1G). The results of soluble sugar content indicated that treatment with 5.0 mM trehalose significantly increased soluble sugar content at 12 h, 24 h, and 36 h, while treatment with 10.0 mM trehalose significantly increased soluble sugar content at 12 h, 36 h, and 48 h under heat stress (Figure 1H).

In summary, considering the morphological phenotypes of tea plants and these physiological indicators, our results suggest that 5.0 mM trehalose could enhance heat-stress tolerance in tea plants under 24 h high-temperature treatment. Consequently, we selected the CK (high temperature 0 h + water), T (high temperature 24 h + water), and TT (high temperature 24 h + 5.0 mM trehalose treatment) samples for subsequent transcriptome analysis to reveal the effect of exogenous trehalose on resistance to heat stress in tea plants.

### 2.3. Transcriptome Analysis and Differentially Expressed Genes Identification for Heat Stress Response after Exogenous Trehalose Treatment in Tea Plant

To further explore the molecular mechanism underlying the response of tea plants to high temperatures in the presence of exogenous trehalose, RNA-seq analysis was performed on the CK, T, and TT samples. An average of 121.52, 110.87, and 111.24 M clean reads were obtained for CK, T, and TT samples, respectively. The Q30 percentage for each library (sequences with sequencing error rate of less than 0.1%) was above 91%, and the average GC content of all libraries was 44.48%. A total of 73.59 to 80.95% of clean reads in each library were successfully mapped to the reference genome (Table 1), indicating that RNA-seq data are of high quality and can be used for further analysis.

Subsequently, a principal component analysis (PCA) was conducted using the quantitative gene expression results for evaluating the samples replication. The results showed that replicates of each treatment clustered closely, and the CK, T, and TT samples were distinctly separated from each other, demonstrating the reliability of the three biological samples (Figure 2A). Further, differentially expressed analysis was performed using DESeq R package (v.1.18.0) to compare the differentially expressed genes (DEGs) in different groups (T vs. CK, TT vs. CK, and TT vs. T). The results revealed that 527 and 470 upregulated DEGs, as well as 1877 and 2793 downregulated DEGs in T vs. CK and TT vs. CK, respectively (Figure 2B). In the TT vs. T comparison group, 508 genes were significantly upregulated, and 1099 genes were downregulated. Among these identified DEGs, only 301 out of 4561 DEGs were common in all three samples, with 1184 DEGs specifically expressed in the TT vs. CK (Figure 2C), indicating that exogenous trehalose can induce changes in the gene expression in tea plants under heat stress.

### 2.4. Gene Ontology and Kyoto Encyclopedia of Genes and Genomes Pathway Analyses of DEGs in C. sinensis under Heat Stress Regulated by Exogenous Trehalose

To investigate the mechanism by which exogenous trehalose alleviates heat stress in tea plants, Gene Ontology (GO) and Kyoto Encyclopedia of Genes and Genomes (KEGG) enrichment analyses were performed on DEGs. As shown in Figure 2D–F, the GO analysis revealed that “response to chitin” and “defense response” were commonly enriched in all three comparison groups (T vs. CK, TT vs. CK, and TT vs. T) in the biological process category. Specifically, “response to salicylic acid”, “hydrogen peroxide catabolic process”, “cell wall organization”, “chitin catabolic process”, and “polysaccharide catabolic process” were uniquely enriched in T vs. CK, TT vs. CK, and TT vs. T, respectively. In the cellular component category, the top four enriched terms of DEGs were consistent in all three comparison groups, including “cell wall”, “apoplast”, “extracellular region”, and “integral component of membrane”. For the molecular function category, “heme binding” was commonly enriched in all three comparison groups, while specific functions such as “carboxylic ester hydrolase activity” and “hydrolase activity, hydrolyzing O-glycosyl compounds” were identified in the T vs. CK group, and “xyloglucan:xyloglucosyl transferase activity” was unique to the TT vs. CK group, and “chitin binding”, “chitinase activity”, and “monooxygenase activity” were found only in the TT vs. T group.

For the KEGG analysis of DEGs in the T vs. CK, TT vs. CK, and TT vs. T comparison groups, results are shown in Figure 2G–I. Interestingly, we found that among the top five enriched pathways, including “Plant-pathogen interaction”, “MAPK signaling pathway-plant”, “Plant hormone signal transduction”, and “Phenylpropanoid biosynthesis”, were conserved across all three comparison groups (Figure 2G–I). Additionally, “Flavone and flavonol biosynthesis”, “Flavonoid biosynthesis”, and “Cutin, suberine and wax biosynthesis” were specifically enriched in the T vs. CK, TT vs. CK, and TT vs. T group, respectively. Notably, galactose metabolism was among the top 20 enriched pathways in TT vs. T, suggesting a potential role of galactose in trehalose-induced tolerance to heat stress in tea plants.

### 2.5. Analysis of DEGs Involved in MAPK Signaling and Plant Hormone Signal Transduction Pathways in Tea Plants under Heat Stress with Exogenous Trehalose Treatment

To investigate the influence of trehalose on the gene expression profiles involved in the MAPK signaling pathway in tea plants under heat stress, the relevant DEGs in this pathway were identified (Figure 3A). A total of 62 DEGs were identified in this pathway. Among them, *OXI1* (*Oxygen Sensor1*), *RTE1* (*REVERSION-TO-ETHYLENE SENSITIVITY 1*), *RAN1* (*Ras-like nuclear protein 1*), *MAPKKK17* (*Mitogen-activated protein kinase kinase kinase kinases 17*), *MAPKKK18* (*Mitogen-activated protein kinase kinase kinase kinases 18*), *MKK2* (*Mitogen-activated protein kinase kinase 2*), *MPK4* (*Mitogen-activated protein kinase 4*), and *EBF1/2* (*EIN3-binding F-box 1/2*) each contained only one member (Figure 3A). The sixteen structural genes, including *FLS2* (*flavonol synthase 2*), *BAK1* (*BRI1-associated receptor kinase 1*), *ETR* (*Ethylene receptor*), *PYR/PYL* (*Pyrabactin resistance*), *PP2C* (*PP2C-type protein phosphatase*), *CaM4* (*Calmodulin 4*), *MPK3/6* (*Mitogen-activated protein kinase 3/6*), *ACS6* (*ACC synthase 6*), *MKS1* (*MKS transition zone complex subunit 1*), *WRKY33* (*WRKYGOK 33*), *WRKY22* (*WRKYGOK 22*), *PR1* (*pathogenesis-related protein 1*), *ERF1* (*ethylene responsive factor 1*), *MYC2* (*Myelocytomatosis proteins 2*), *RbohD* (*Respiratory burst oxidase homolog protein D*), and *CHIB* (*chitinase ChiB*) contained more than two members (Figure 3A). Analysis of the transcript levels of the DEGs revealed that most DEGs had the highest expression levels in the CK group, such as *FLS2*, *BAK1*, *RTE1*, *ETR*, *RAN1*, *PYR/PYL*, *CaM4*, *MAPKKK18*, *MKK2*, *MPK3/6*, *ACS6*, *MKS1*, *WRKY33*, *WRKY22*, *EBF1/2*, etc. In the comparison between T vs. CK groups, the expression levels of two members of the *MPK3/6*, one member of the *ERF1*, and four members of the *CHIB* in the ethylene pathway were significantly increased in the T sample, suggesting that the ethylene signaling pathway was involved in defending against heat-stress-induced injury. In the TT vs. T group, the expression level of one member of the *PYR/PYL* family and three members of the *PP2C* family was significantly increased in TT, while the expression of *RbohD* in TT decreased, suggesting that trehalose treatment induced the ABA and ROS signaling pathway in response to heat stress.

The DEGs in the ABA signal transduction pathway were analyzed with qRT-PCR. The results showed that the relative expressions of *CsPP2C3* and *CsPYL4* under heat stress in the TT sample were significantly higher than in the T samples, while the relative expressions of *CsMAPKKK17/18* in the T sample were significantly higher than those in the TT sample (Figure 3B). In addition, the expression levels of these DEGs via qRT-PCR analysis was consistent with transcriptome data.

A total of 68 DEGs were identified in plant hormone signal transduction, as shown in Figure 3C. Among these DEGs, 11 structural genes contained more than two members, including *XTH* (*xyloglucan endotransglucosylase/hydrolase*), *SAUR* (*Small auxin-up RNA*), *PYL*, *ERF*, *AUX/IAA* (*Auxin/Indole-3-acetic acid*), *IAA* (*indole-3-acetic acid*), *GH3* (*Gretchen Hagen 3*), *MYC*, *GAI* (*Gibberellic Acid Insensitive*), *AUX* (*Auxin*), and *PP2C*. Other structure genes like *ARR* (*Arabidopsis response regulators*), *ARF* (*Auxin Response Factor*), *EBF*, *and AHP* (*Histidine-phosphate transporter*) contained one member each. Analysis of the expression abundance of 68 DEGs revealed that most genes had higher expression abundance in the CK group. Notably, the expression of four auxin early-response members and one *ERF1* gene member in the T group was higher than that in the CK group, indicating that the auxin pathway and ethylene pathway may be responsible for heat stress in tea plants. In the TT vs. T comparison group, the expression of three *PP2C* gene members and one *PYL* was increased, which was consistent with the identification results of the MAPK signal transduction pathway, indicating that trehalose-mediated heat stress may induce the response of the ABA signaling pathway in tea plants.

The DEGs of the IAA signal transduction pathway were also analyzed with qRT-PCR (Figure 3D). The results showed that the relative expression of three *CsAUX1 (AUXIN RESISTANT 1*) in the CK samples was the highest, while the relative expression of *CsAUX1-1* and *CsAUX-3* in the T samples was significantly higher than that in the TT samples. *CsGH3.1* in the T group was significantly higher than in the CK and TT samples. Furthermore, the relative expressions of *CsIAA-1* and *CsIAA-2* in the CK and T samples were significantly higher than those in the TT sample, whereas *CsIAA16-3* in the TT samples was significantly higher than that in the T sample. The expression level of *CsSAUR50* was the highest in the T sample, and *CsSAUR36* had the highest expression in the CK samples. Therefore, trehalose may play a negative regulatory role in IAA signal transduction under heat stress.

### 2.6. Analysis of DEGs Involved in Phenylpropanoid Biosynthesis, Flavonoid Biosynthesis, and Flavone and Flavonol Biosynthesis in Tea Plants under Heat Stress with Exogenous Trehalose Treatment

A total of nine classes of structure genes assigned to phenylpropanoid biosynthesis were identified, including *PAL* (*Phentlalanin ammonialyase*), *4CL* (*4-coumaric acid coenzyme A ligase*), *HCT* (*hydroxycinnamoyl-CoA: shikimate/quinate hydroxycin-namoyltransferase*), *CCoAMT* (*trans-caffeoyl-coenzyme A 3-O-methyltransferase*), *F5H* (*Ferulate 5-hydroxylase*), *ALD* (*Acetaldehyde dehydrogenase*), *CAD* (*cinnamyl alcohol dehydrogenase*), *POD* (*Peroxidase*), and *UGT* (*uridine diphosphate glycosyltransferase*). Twenty-two DEGs were identified in the flavonoid biosynthesis, including *HCT*, *CCoAMT*, *CHS* (*chalcone synthase*), *PTG1* (*poly cistronic tRNA-sgRNA*), *FLS* (*Flavonol Synthase*), *CYP75A* (*flavonoid 3′,5′-hydroxylase*), *CYP75B* (*flavonoid 3′-hydroxylase*), *DFR* (*dihydroflavonol-4-reductase*), *ANS* (*Anthocyanidin synthase*), *LAR* (*leucoanthocyantin reductase*), and *ANR* (*Anthocyanidin reductase*). Three DEGs were identified in flavone and flavonol biosynthesis, which were *F3GT1* (*flavonoid 3-O-glucosyltransferas*), *CYP75A*, and *CYP75B*. Since phenylpropanoid biosynthesis, flavonoid biosynthesis, and flavone and flavonol biosynthesis are interconnected, the above pathways were mapped based on the identified DEGs (Figure 4A). Further analysis of the expression patterns of DEGs in phenylpropanoid biosynthesis showed that the majority of DEGs were expressed at higher levels in the CK sample than in the T and TT samples, such as *PAL*, *4CL*, *CCoAMT*, and *UGT*, which are involved in lignin biosynthesis, with individual genes upregulated in the T or TT sample. Specifically, a member of HCT was upregulated in the T and TT samples, and *F5H* was upregulated in the T samples. Analysis of the expression patterns of DEGs in flavonoid biosynthesis and flavone and flavonol biosynthesis showed that gene expression levels were lower in both T and TT samples than in the CK sample, indicating that high temperature may inhibit the accumulation of tea flavonoids. Additionally, one *CYP75A* gene and one *HCT* member in the TT vs. T comparison group were upregulated in the TT samples separately, possibly due to glucose induction.

Four DEGs biosynthesized with lignin and flavonoids were selected for qRT-PCR analysis (Figure 4B). The results showed that the relative expression levels of these four genes in TT samples were all higher than that of T, suggesting that trehalose may induce changes in lignin biosynthesis and flavonoid metabolites through regulation of the expression of these genes.

### 2.7. Analysis of DEGs Involved in Galactose Metabolism in Tea Plants under Heat Stress with Exogenous Trehalose Treatment

As shown in Figure 4C, galactose metabolism is a specifically enriched metabolic pathway in the TT vs. T comparison group, which includes genes encoding *HXK* (*hexokinase*), *GOLS* (*galactinol synthase*), *RFS5* (*raffinose synthase*), and *CWINV* (*cell-wall invertase*). Among them, five *GOLS* and one *RFS* have the highest expression in the TT samples, while the expression of two *CWINV* genes shows a decreasing trend. These results suggest that trehalose may help tea plants withstand heat stress by upregulating the expression of genes involved in the galactose metabolic pathway.

The DEGs of the galactose metabolism pathway were further analyzed using qRT-PCR (Figure 4D). The results revealed that the relative expression level of *CsGOLS2* in the TT samples was significantly higher than that in the CK and T samples, the expression level of *CsRAFS5* in the TT samples was notably higher than that in the T samples, and the expression level of *CsCWINV1* was highest in the T samples.

### 2.8. Analysis of Transcription Factors in Response to Heat Stress under Exogenous Trehalose Treatment in Tea Plants

To explore the effect of trehalose on the expression of transcription factors in tea plants under high-temperature stress, the expression levels of various transcription factors, including *ARF* (*auxin response factor*), *bHLH* (*basic helix-loop-helix*), *NAC* (*NAM*, *ATAF1/2* and *CUC2*), *HSF* (*Heat Shock Factors*), *HSPs* (*Heat Shock Proteins*), *MYBs* (*myeloblastosis*), *WRKYs* (*WRKYGQK*), and *ERF*, were analyzed. The results showed differential expression patterns, with 4, 20, 23, 2, 10, 37, 29, and 46 transcription factors showing altered expression levels of *ARF*, *bHLH*, *NAC*, *HSF*, *HSP*, *MYB*, *WRKY*, and *ERF*, respectively.

Subsequently, heat map analysis of these differentially expressed transcription factors showed that most members of *MYB*, *ARF*, *ERF*, *NAC*, *bHLH*, and *WRKY* were downregulated in both T and TT groups compared to the CK group (Figure 5A–H), potentially contributing to the higher number of downregulated genes in the different comparison groups. Notably, *HSFB2B*, a member of the *HSF* family, was significantly upregulated in both T and TT samples, with the highest expression in TT samples. Furthermore, some members of *HSPs*, which are directly regulated by *HSF*, were significantly upregulated in the TT vs. CK comparison group, with two showing higher transcript levels in the TT vs. T samples. Therefore, *CsHSFB2B*, *CsHSP18.1*, and *CsHSP26.5* were further selected for qRT-PCR analysis, and the results confirmed that the relative expression levels of the three genes were significantly higher in the TT samples than that in T samples (Figure 5I). These findings suggested that trehalose may induce the expression of *HSF* to improve the heat resistance of tea plants.

## 3. Discussion

### 3.1. Trehalose Alleviates the Damage Caused by Heat Stress through Increasing the Activity of Antioxidant Enzymes and the Content of Osmotic Substance

The morphological phenotypes of tea plants and the measured physiological indicators demonstrated that 5.0 mM trehalose treatment could significantly improve the stress tolerance of tea plants after 24 h of heat stress. The application of exogenous 5.0 mM trehalose reduced the degree of wilting of tea leaves under heat stress for 24 h to some extent (Figure 1B). This improvement can be attributed to several factors. Firstly, trehalose can increase the activity of antioxidant enzymes (Figure 1C,D) and reduce excessive ROS content [22]. Secondly, trehalose can decrease the MDA content of tea plants under heat stress (Figure 1E), thereby mitigating cell membrane damage. It is worth noting that the decline in MDA content is likely a reflection of the protective effect against heat stress rather than a direct cause of reduced heat stress [23]. Thirdly, trehalose can increase the contents of PRO (Figure 1F), endogenous trehalose, and soluble sugar (Figure 1G,H) in tea plants subjected to heat stress, thereby enhancing the stress resistance [15,24,25]. These findings align with previous studies showing that trehalose can improve plant heat tolerance by enhancing antioxidant enzymes and osmotic substances [21,26].

### 3.2. Trehalose May Enhance the Heat Tolerance of Tea Plants by Activating the ABA and MAPK Signal Transduction Pathway

Plant hormones serve as crucial regulators in plant responses to adversity, playing a pivotal role in activating the defense mechanism of plants against various stresses [27]. Among these hormones, ABA is considered a “stress hormone” that is particularly important in response to heat stress [28,29]. Typically, under abiotic stress conditions, the ABA receptor protein PYL inhibits the phosphatase activity of PP2C, releasing SnRK2 and transmitting ABA signals by phosphorylating downstream genes [30,31]. The *PYL* and *PP2C* gene families consist of numerous members, and their interactions are very complex. PYLs exhibit selective inhibition of PP2Cs, with different degrees of inhibition [32]. Zhang et al. proved that *CsPP2C* was identified as *HAI3* with a special structure in the *PP2C* subfamily of group A [33]. In our study, the expression of *CsPYL4* and *CsPP2C3* tends to be upregulated (Figure 3). This could be due to the reason that *CsPYL4* has no inhibitory effect on *HAI3*, or its inhibitory activity is weak. Alternatively, trehalose may induce the regulation of *PP2C* expression, although its exact mechanism remains unknown. Numerous studies have demonstrated that the expression of *PYL* and *PP2C* can enhance plant resistance to abiotic stress [34,35]. In addition, it has been reported that the class *PP2C* responds to osmotic stress by inhibiting *MAPKKK* [36]. Our research also demonstrated that trehalose leads to the downregulation of MAPK signal pathway genes, such as *CsMAPKKK17* and *CsMAPKKK18*, downregulated by *CsPP2C3*. Among this pathway, a total of 62 DEGs were identified, and most of them (*MAPKKK17*, *MAPKKK18*, *MKK2*, *MPK4*, and *EBF1/2*) exhibited their highest expression levels in the CK group; only two members of *MPK3/6* showed increased expression levels in the T group (Figure 3A). In conclusion, it is hypothesized that trehalose may induce the transcriptional activation of *CsPYL4* and *CsPP2C3* in the ABA signaling pathway, as well as the transcriptional inhibition of *CsMAPKKK17* and *CsMAPKKK18* via *CsPP2C3*, thereby improving the heat-stress tolerance of tea plants.

### 3.3. Trehalose Could Induce Gene Expression in the Galactose Pathway of Tea Plants under Heat Stress and Improve the Heat Resistance of Tea Plants

Sugars are essential for plant resistance to stress as they act as osmoregulatory substances that help maintain cellular homeostasis [37,38]. According to Figure 4, the galactose metabolic pathway was significantly enriched in the TT vs. T comparison group, with five *CsGOLS* and one *CsRFS5* showing significantly upregulation. *GOLS* and *RFS* are key genes involved in cottonseed sugar synthesis and are known to respond to abiotic stresses [39,40,41]. Previous studies have demonstrated that *GOLS* and *RFS* genes are induced by heat shock transcription factor HSF to enhance stress resistance in transgenic plants under high-temperature treatment [42,43]. In our study, *CsGOLS2* and *CsRFS5* were significantly upregulated in trehalose-treated samples, indicating that trehalose can induce the expression of key synthases in the raffinose synthesis pathway under high-temperature stress to combat heat stress. Additionally, trehalose may induce the upregulation of HSF, indirectly leading to increased expression of *CsGOLS2* and *CsRFS5*.

### 3.4. Trehalose Enhances the Heat Resistance of Tea Plants by Inducing the Expression of Key Genes Involved in Lignin Synthesis in the Phenylpropanoid Pathway

The lignin biosynthesis pathway is a component of the broader phenylpropanoid pathway, leading to the synthesis of lignin, which plays a crucial role in safeguarding cell walls during plant development or in response to external stimuli [44,45]. POD is a key enzyme in the final step of catalytic synthesis within the lignin metabolic pathway [44,46]. In our study, three *CsPOD* genes were upregulated, and ten *CsPOD* genes were downregulated in the TT group compared to T group (Figure 4). Additionally, *CsHCT* was upregulated in the TT samples compared to the T and CK groups. *CsHCT*, a montmorillonite/quinic acid hydroxycinnamoyl transferase in the phenylpropanoid metabolic pathway, directly controls the biosynthesis of lignin monomers and plays an important role in lignin synthesis [47]. In the TT vs. T comparison group, the effect of trehalose on flavonoid synthesis was observed, with one *CsHCT* and one *CsCYP75A* being upregulated. It is reported that CYP75A is a key enzyme involved in the catechins formation, known for its strong radical scavenging and antioxidant effects [48]. In conclusion, exogenous trehalose induced the changes in the expression of *PODs* in the process of lignin synthesis process and secondary metabolite synthesis in response to high-temperature damage in tea plants.

### 3.5. Trehalose Improves the Heat Tolerance of the Tea Plant by Inducing the Expression of HSF and HSP under Heat Stress

HSF plays an important role in basic heat tolerance in plants and can regulate the accumulation of HSP [49]. In this study, a heat shock transcription factor *CsHSFB2B*, along with two heat shock proteins, *CsHSP18.1* and *CsHSP26.5*, were found to be upregulated in the TT samples. *CsHSFB2B* belongs to the HSFB subfamily and serves as a transcription inhibitor. Ikeda et al. demonstrated that the heat tolerance of a *HSFB1* and *HSFB2b* double mutant in *Arabidopsis thaliana* at 42 °C surpasses that of the wild type, with the heat-induced *HSF* expression of *HSFB1* and *HSFB2b* remaining higher than that in the wild type [50]. HSFB1 and HSFB2B are also essential for the expression of heat-stress-inducible HSP, and the activation and repression of HSFs transcription factors plays a crucial role in the heat-stress response. Kolmos et al. found that the *hsfb2b* mutant exhibits a shortened circadian rhythm under high-temperature stress [51]. The circadian rhythm is an activator of abiotic stress signals and plays an important role in plant growth and adaptation. In our study, we identified two small molecule HSP proteins CsHSP18.1 and CsHSP26.5 in response to heat stress. It has been reported that the expression of CsHSP18.1 protein gradually increased with rising temperatures in tea plants [52]. Vianna et al. found that the small heat shock protein Hsp26 is essential for the high heat-stress tolerance of the fil1 mutant, both in stationary phase cells and active fermentation [53]. In conclusion, exogenous trehalose has the potential to induce the expression of *CsHSFB2B* in response to heat stress, leading to the upregulation the transcription levels of *CsHSP18.1* and *CsHSP26.5* in tea plant under heat stress conditions.

Noteworthily, there is a discrepancy in gene expression levels observed between Figure 4 and Figure 5. This may indeed be attributed to the distinct roles played by transcription factor genes and structural genes in the plant’s response to stress and treatments. In Figure 5, where the focus is on transcription factor genes, the expression levels are high in the treatment group, reflecting the rapid regulatory responses activated by these genes in response to the applied treatment. On the other hand, in Figure 4, which features structural genes, one gene shows low expression in the treatment group, indicating potential downstream effects that may occur as a result of the regulatory events initiated by the transcription factors. This distinction underscores the complexity of gene regulatory networks and the interconnected roles of different gene categories in shaping the plant’s response to external stimuli. By considering these factors, we can gain deeper insights into the molecular mechanisms underlying stress responses and treatment effects in plants.

## 4. Materials and Methods

### 4.1. Plant Materials and Heat Stress Treatment

Two-year-old tea plants (*C. sinensis* cv. Tieguanyin) were placed in an artificial climate incubator with a photoperiod of 12 h light (25 °C)/12 h darkness (19 °C) and 75% relative humidity for 7 days. Subsequently, the tea plants were randomly divided into four groups and subjected to heat-stress treatment under a 12 h light (38 °C)/12 h darkness (29 °C) cycle, maintaining 75% relative humidity, as reported by previous studies [54,55,56]. After 2 days, the tea leaves were sprayed with water, 2.5 mM, 5.0 mM, and 10.0 mM trehalose, respectively. The plants continued to be cultured under high temperatures (38 °C/29 °C, 12 h light/12 h darkness) as shown in Figure 1A. For each treatment, leaves from the second and third positions were collected at 0, 12, 24, 36, and 48 h, respectively. Each sample was collected with three biological replicates from the same part of the plant to generate an individual sample, and then immediately frozen using liquid nitrogen stored at −80 °C refrigerator for subsequent physiological analysis and transcriptome sequencing.

### 4.2. Determination of Antioxidant Enzyme Activity

The activities of superoxide dismutase (SOD) and peroxidase (POD) were determined based on the reference kit’s instructions (Suzhou Grace Biotechnology Co., Ltd., Suzhou, China). A total of 0.1 g of leaves were thoroughly ground on ice using the reaction buffer from the kit, and the resulting homogenate was then transferred to a 1.5 mL centrifuge tube. Following centrifugation at 4 °C and 12,000 rpm for 10 min, the supernatant was collected and kept on ice for measurement. The SOD and POD activities were determined at 450 nm and 470 nm, respectively, as described in the protocol [57].

### 4.3. Determination of MDA, Proline, Soluble Sugar, and Trehalose Contents

The MDA content was determined via the thiobarbituric acid method [58]. In brief, 0.5 g of fresh leaves were mixed with 2 mL of 0.05 mol L^−1^ phosphate buffer (pH 7.8) and homogenized in an ice bath. The homogenate was then centrifuged at 4500 rpm for 10 min. Subsequently, 2 mL of the supernatant were mixed with 3 mL of 0.5% thiobarbituric acid and 5.0% trichloroacetic acid solution. The resulting reaction mixture was kept in a boiling water bath for 10 min, followed by rapid cooling and centrifugation at 4500 rpm for 10 min. The content of MDA in the supernatant was determined at 532 nm and 600 nm, respectively.

The content of proline (PRO) was determined based on the instructions for the reference kit (Suzhou Grace Biotechnology Co., Ltd.). According to the protocol, 0.1 g of leaves were homogenized with 1 mL of sulfosalicylic acid in an ice bath. The homogenate was then transferred to a 1.5 mL EP tube and heated in a water bath at 90 °C for 10 min. After centrifugation at 25 °C and 12,000 rpm for 10 min, the PRO content was determined by calculating the absorbance at 520 nm based on a PRO standard curve [59].

The soluble sugar content was measured using the anthrone method [60]. Briefly, 0.1 g leaves were ground with 1 mL of 80% ethanol in an ice bath. The resulting slurry was transferred to a 1.5 mL centrifuge tube and made up to 1.5 mL with the 80% ethanol. Afterward, the mixture was incubated in a 50 °C water bath for 20 min, followed by cooling to room temperature. After centrifugation at 12,000 rpm at room temperature for 10 min, the soluble sugar content was determined at 620 nm and quantified based on a soluble sugar standard curve.

The content of endogenous trehalose was determined using the anthrone-sulfuric acid method [61]. A total of 0.05 g of tea leaves were ground in a 1 mL reaction buffer, shaken at room temperature for 30 min, and then centrifuged at 8000 rpm for 10 min at room temperature. Finally, the absorbance of the reaction mixtures at 620 nm was measured, and the trehalose level was calculated based on a trehalose standard curve.

### 4.4. RNA Extraction and Transcriptome Sequencing

Total RNA was extracted using the Ambion-1561 mirVana ^TM^ miRNA Isolation Kit (ThermoFisher Scientific, Waltham, MA, USA) according to the manufacturer’s protocol. RNA purity and quantification were measured using a NanoDrop 2000 spectrophotometer (ThermoFisher Scientific). RNA integrity was determined via 1.0% agarose gel electrophoresis, and only samples meeting the following criteria were used for library construction: RIN (representing RNA Integrity Number) value ≥ 7, 28S/18S ≥ 0.7, OD260/280 value between 1.8~2.2, and a concentration of ≥50 ng/μL. The Ribo-off rRNA Depletion Kit (Vazyme, Nanjing, China) was employed to eliminate ribosomal RNA, and then the libraries were constructed using VAHTS Universal V6 RNA-seq Library Prep Kit according to the manufacturer’s instructions. The transcriptome sequencing and analysis were conducted by OE Biotech Co., Ltd. (Shanghai, China).

### 4.5. RNA-Seq Data Analysis

To ensure the accuracy and reliability of the sequencing results, the raw data after sequencing was filtered to remove reads containing ribosomal RNA (rRNA), adapter, fuzzy base (N), and low-quality base using Trimmomatic software (v.0.39) [62]. Then, the quality of reads was detected with fastp [63] to ensure their accuracy. Finally, the reads that passed stringent filtering criteria were designated as “Clean reads” for subsequent analysis. Hisat2 was used to align Clean reads to the reference genome of the Tieguanyin tea plant [33,64], and the alignment results were evaluated using RseqQC [65]. String Tie was used to assemble single transcripts from each sample aligned to the reference genome and spliced into Merged Transcripts.

The sequence similarity alignment method was used to determine the abundance levels of protein-coding gene expression. The htseq-count (HTSeq 2.0) [66] was used to obtain the number of Clean reads aligned to the protein-coding genes. The FPKM (Fragments per kilobase per million mapped reads) value was calculated with Cufflinks [67] to normalize the expression abundance of protein-coding genes.

### 4.6. Different Expression Analysis

The DESeq R package (v.1.18.0) [68] was applied to normalize the gene counts of each sample, calculate the fold changes, and use the negative binomial distribution check to test the significance of differences in reads. Finally, the differentially expressed genes (DEGs) were identified according to the screening criteria of |log_2_FC| > 1 and *p*-value < 0.05.

The sequences of the screened DEGs were aligned to Swiss-Prot to obtain Gene Ontology (GO) and Kyoto Encyclopedia of Genes and Genomes (KEGG) annotations for each mRNA transcript, and GO and KEGG enrichment analyses were performed to uncover the biological functions and metabolic pathways influenced by the DEGs. GO analysis detects the enrichment of DEGs in particular cellular components (CC), biological processes (BP), and molecular functions (MF) [69]. KEGG enrichment analysis reveals which pathways DEGs are involved in. In this study, GO and KEGG enrichment analyses were both performed using the ClusterProfiler package (version 3.16.0) in R (with a *q* < 0.05 and a *p* < 0.05 considered significant) [70].

### 4.7. Quantitative Real-Time PCR (qRT-PCR) Analysis

Real-time quantitative PCR (qRT-PCR) was used to determine the gene expression level. *GAPDH* and *β-actin* were used as internal reference genes, and the relative expression was calculated using the 2^−ΔΔCT^ method [71]. All gene expression analyses were performed across three independent biological replicates, and all data were presented as mean ± standard deviation (SD).

### 4.8. Statistical Analysis

SPSS 25.0 software and Microsoft Excel 2016 software were used to perform the statistical analyses. Differences among groups were assessed using a one-way analysis of variance (ANOVA) with Duncan’s post hoc test. *p* < 0.05 was considered to be significant.

## 5. Conclusions

In conclusion, our study demonstrates that the application of exogenous trehalose enhances the heat resistance of tea plants through various mechanisms. It increases the activity of antioxidant enzymes, boosts osmotic substances, and reduces MDA content. Consistent with these physiological changes, transcriptome analysis revealed that trehalose could regulate the expression of genes (*CsPYL4*, *CsPP2C3*, *CsMAPKKK17*, and *CsMAPKKK18*) involved in the ABA signaling pathway to defend against heat stress. It also improved antioxidative ability by upregulating the expression of *CsGOLS* and *CsRFS5* in the raffinose synthesis pathway and enhanced the ability to scavenge free radicals by upregulating the expression of *CsHCT* and *CsCYP75* in the lignin and flavonoid synthesis pathway, as well as enhancing the heat resistance by increasing the expression of *CsHSFB2B*, *CsHSP26.5*, and *CsHSP18.1* (Figure 6). Overall, exogenous trehalose can effectively enhance the heat resistance of tea plants, making it a promising agent for improving the heat tolerance of tea plants.

## Figures and Tables

**Figure 1 plants-13-01339-f001:**
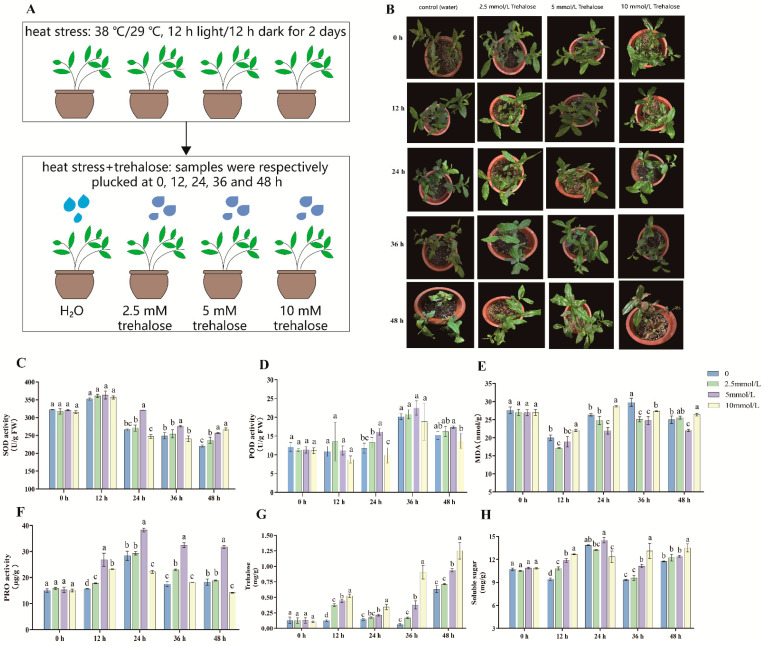
Effects of exogenous trehalose on growth morphology and physiological characteristics of tea plants under heat stress. (**A**): The trehalose treatment process for tea plant under heat stress. (**B**): Morphological phenotypes of tea plants treated with different concentration of exogenous trehalose under heat stress. (**C**–**H**): SOD activity (**C**), POD activity (**D**), MDA (**E**), PRO (**F**), endogenous trehalose (**G**), and soluble sugar (**H**) contents in tea leaves at 0, 12, 24, 36, and 48 h after treatment with exogenous trehalose under heat stress. Error bars represent mean ± SD, n = 3. Lowercase letters indicate the significance of *p* < 0.05 between different concentrations of trehalose treatment.

**Figure 2 plants-13-01339-f002:**
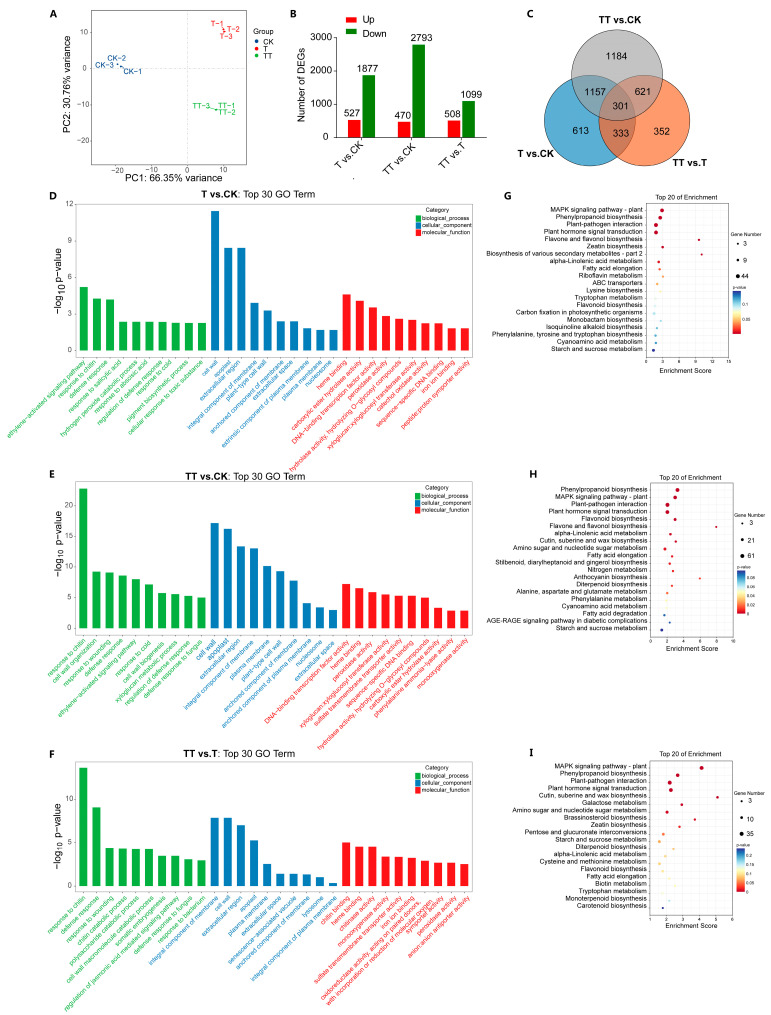
RNA-seq analysis and DEGs identification in tea plants under heat stress with exogenous trehalose treatment. (**A**): PCA analysis of RNA-Seq data in tea plants from CK, T, and TT samples. (**B**): Number of DEGs in the T vs. CK, TT vs. CK, and TT vs. T groups. (**C**): Venn diagrams of DEGs in the T vs. CK, TT vs. CK, and TT vs. T groups. (**D**–**F**): GO enrichment of the DEGs in the T vs. CK, TT vs. CK, and TT vs. T, respectively. (**G**–**I**): KEGG enrichment of the DEGs in the T vs. CK, TT vs. CK, and TT vs. T, respectively.

**Figure 3 plants-13-01339-f003:**
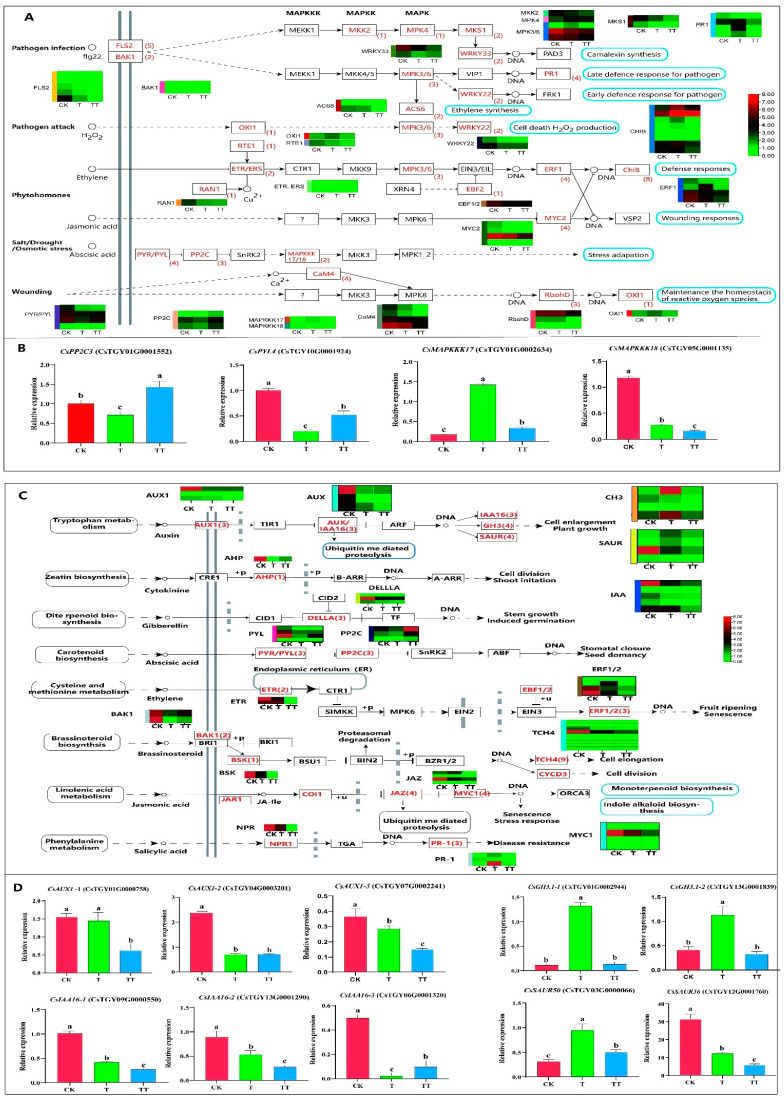
DEGs analysis related to the MAPK signaling and plant hormone signal transduction pathways in tea plants under heat stress with exogenous trehalose treatment. (**A**): Analysis of DEGs in the MAPK signal transduction pathway. (**B**): The qRT-PCR analysis of DEGs in ABA signal transduction. (**C**): Analysis of DEGs in plant hormone signal transduction. (**D**): The qRT-PCR analysis of DEGs in IAA signal transduction. The error bar represents the mean ± standard deviation, n = 3. Lowercase letters showed significant differences (*p* < 0.05).

**Figure 4 plants-13-01339-f004:**
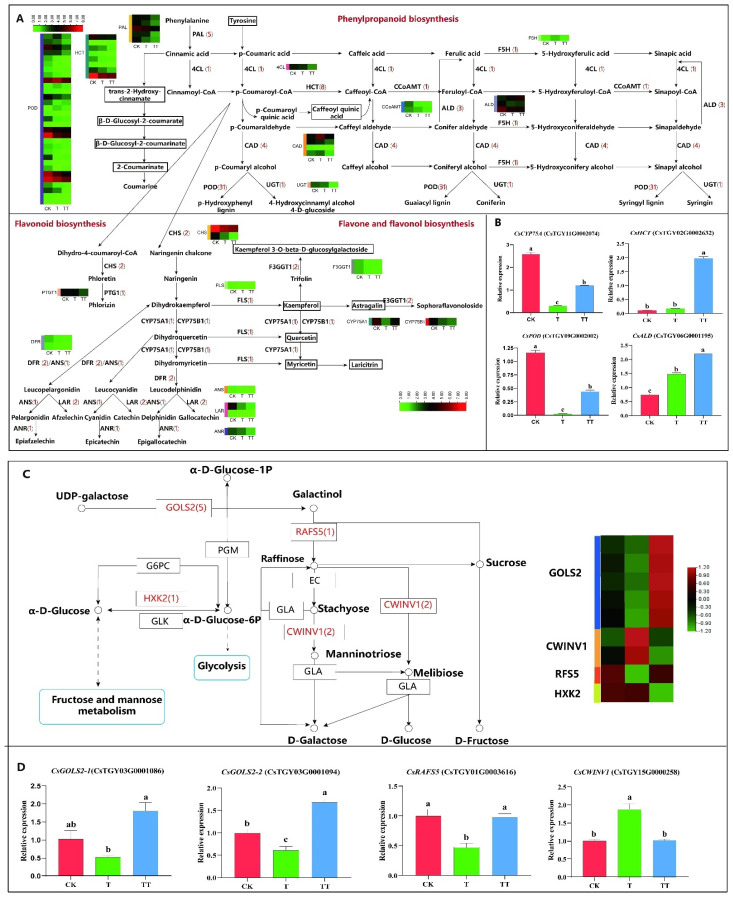
Analysis of DEGs involved in different biosynthetic and metabolic pathways in tea plants under heat stress with exogenous trehalose treatment. (**A**): Analysis of DEGs in plant hormone signal transduction. (**B**): The qRT-PCR analysis of DEGs in IAA signal transduction. (**C**): Analysis of DEGs in galactose metabolism. (**D**): The qRT-PCR analysis of DEGs in galactose metabolism. The error bar represents the mean ± standard deviation, n = 3. Lowercase letters show significant differences (*p* < 0.05).

**Figure 5 plants-13-01339-f005:**
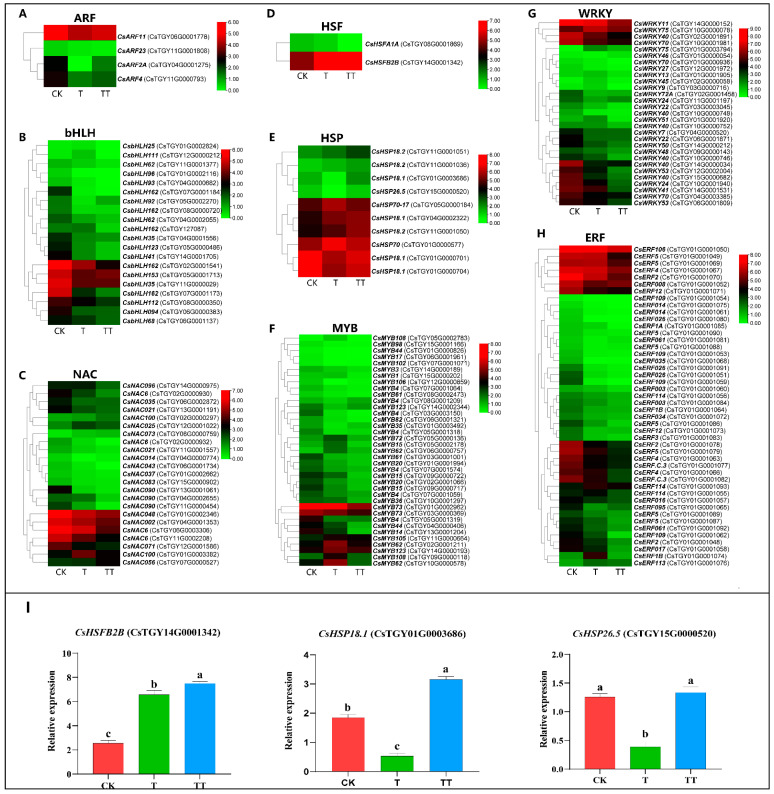
Analysis of transcription factors in response to heat stress under exogenous trehalose treatment in tea plants. (**A**–**H**): The expression heat map transcription factors for *ARF*, *bHLH*, *NAC*, *HSF*, *HSP*, *MYB*, *WRKY*, and *ERF*, respectively. (**I**): The qRT-PCR analysis for *CsHSFB2B* and *CsHSP18.1*, respectively. The error bar represents the mean ± standard deviation, n = 3. Lowercase letters showed significant differences (*p* < 0.05).

**Figure 6 plants-13-01339-f006:**
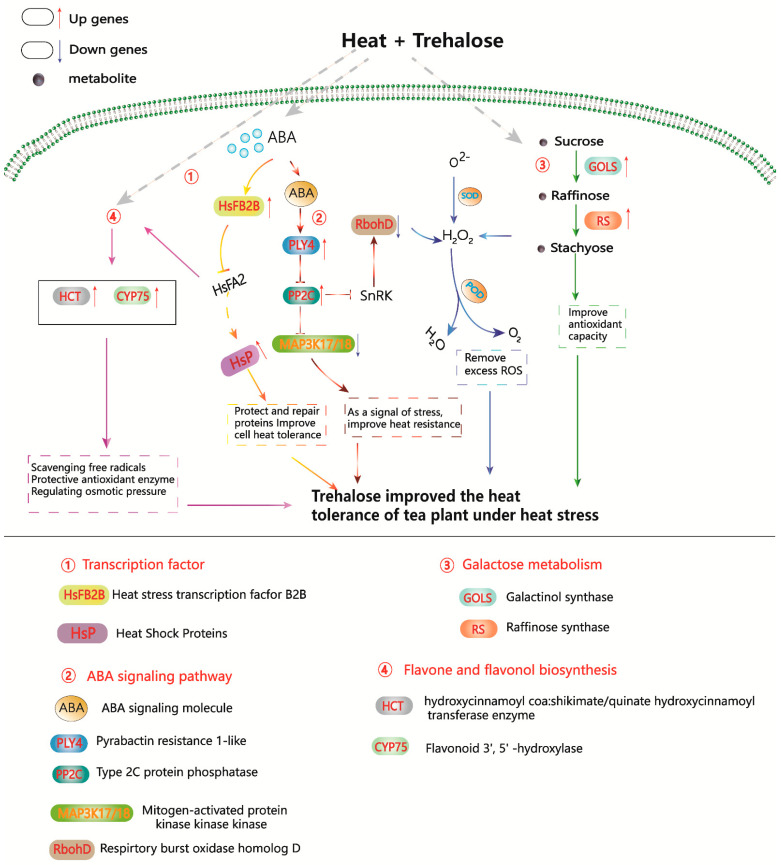
Schematic model of heat-stress response induced by trehalose in tea plants. Red letters are genes and black letters are metabolites. Red arrows indicate upregulated DEGs and bule arrows indicate downregulated DEGs.

**Table 1 plants-13-01339-t001:** Summary of RNA-seq data from Tieguanyin tea plants of CK, T, and TT samples.

Sample	Total Raw Reads (Mb)	Clean Reads (Mb)	Clean Bases (Gb)	Q30(%)	GC (%)	Total Mapped Reads (%)
CK-1	117.79	113.10	16.96	91.52%	44.02%	74.45
CK-2	134.47	130.16	19.52	91.50%	45.71%	76.28
CK-3	126.45	121.31	18.20	91.51%	43.92%	74.58
T-1	117.20	113.68	17.05	91.54%	45.52%	80.95
T-2	119.72	115.70	17.36	91.65%	44.37%	80.28
T-3	107.14	103.24	15.49	91.71%	44.04%	80.17
TT-1	125.50	119.94	17.99	91.54%	44.34%	74.09
TT-2	112.49	107.32	16.10	91.56%	44.14%	73.65
TT-3	111.43	106.45	15.97	92.02%	44.28%	73.59

## Data Availability

Data are contained within the article.

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
