# Peer review of "Physiological and Transcriptome Analyses Reveal the Protective Effect of Exogenous Trehalose in Response to Heat Stress in Tea Plant (Camellia sinensis)"

_plants, 2024, doi:10.3390/plants13101339_

Round 1

Reviewer 1 Report

Comments and Suggestions for Authors

Review Report for Manuscript Ref. [Insert Ref. Number]

Title: Physiological and Transcriptome Analyses Reveal the Protective Effect of Exogenous Trehalose in Response to Heat Stress in Tea Plant (Camellia sinensis)

General Comments: The manuscript presents an insightful study on the protective effect of exogenous trehalose in response to heat stress in Camellia sinensis, highlighting the identification of key genes involved. The article is well-written and of interest to readers. However, there are several concerns that need to be addressed to improve the clarity and comprehensiveness of the study.

Specific Comments:

1.      In the abstract section, it would be beneficial for the authors to mention the activity of antioxidant enzymes under different time intervals (0, 12, 24h, etc.) to provide a more comprehensive understanding of the temporal dynamics of their response.

2.      The manuscript should specify which genes were identified for lignin biosynthesis, as lignin is a crucial metabolite that provides strength to cell walls under abiotic stress conditions.

3.      Was the MAPK pathway the main target in this study? If so, this should be clearly stated in the text.

4.      The introduction section lacks a sufficient number of recent citations. The authors should consider adding more up-to-date references to strengthen their argument.

5.      The manuscript should clearly articulate the gap in earlier studies and emphasize the novel aspects of the current study to highlight its contribution to the field.

6.      Considering the numerous members in TF families, the rationale for selecting CsHSFB2B and CsHSP18.1 for qRT-PCR should be provided to justify their choice over other members.

7.      In Figure 5, the expression of genes is high in the treatment group, but in Figure 4, one gene shows low expression in the treatment group. The authors should discuss the implications of this discrepancy, particularly if the gene's expression is high in the control group.

8.      The discussion section would benefit from more references to strengthen the arguments and provide a more thorough analysis of the results.

9.      The conclusion should be clearer and more concise, summarizing the key findings and their implications effectively.

10.   The materials and methods section could be more descriptive, particularly before section 4.4, to ensure reproducibility of the study.

11.   Some improvements in English language usage and clarity are needed throughout the manuscript.

Overall, the manuscript presents valuable insights into the protective role of exogenous trehalose in heat-stressed Camellia sinensis, but addressing the above concerns will enhance the quality and impact of the study.

Comments on the Quality of English Language

Minor language editing is required

Reviewer 2 Report

Comments and Suggestions for Authors

I don't understand how author's came up with the 38℃/29℃ designated as heat stress give reference of previous research group doing the same or physiological data to show that its designated as heat stress. 

where is the positive control experiment that showing that Exogenous Trehalose levels without heat stress. Meaning that you need to show a time scale experiment from 0h to 48h in control plants showing the variation then only it will be comparable with heat stress.

Fig2: The category fonts are too small that nothing is visible. Enhance the font size.

Give method details for the enrichment analysis. What was the fold enrichment method?

Comments on the Quality of English Language

Nothing major.

Reviewer 3 Report

Comments and Suggestions for Authors
